# Studies of Potency and Efficacy of an Optimized Artemisinin-Quinoline Hybrid against Multiple Stages of the *Plasmodium* Life Cycle

**DOI:** 10.3390/ph14111129

**Published:** 2021-11-06

**Authors:** Helenita C. Quadros, Aysun Çapcı, Lars Herrmann, Sarah D’Alessandro, Diana Fontinha, Raquel Azevedo, Wilmer Villarreal, Nicoletta Basilico, Miguel Prudêncio, Svetlana B. Tsogoeva, Diogo R. M. Moreira

**Affiliations:** 1Instituto Gonçalo Moniz, Fundação Oswaldo Cruz (Fiocruz), Salvador 40296-710, Brazil; helenita_quadros@hotmail.com; 2Organic Chemistry Chair I and Interdisciplinary Center for Molecular Materials (ICMM), Friedrich-Alexander University of Erlangen-Nürnberg, Nikolaus Fiebiger-Straße 10, 91058 Erlangen, Germany; aysun.capci@icloud.com (A.Ç.); lars.herrmann@fau.de (L.H.); 3Dipartimento di Scienze Farmacologiche e Biomolecolari, Università degli Studi di Milano, 20133 Milan, Italy; sarah.dalessandro@unimi.it; 4Instituto de Medicina Molecular, Faculdade de Medicina, Universidade de Lisboa, Av. Prof. Egas Moniz, 1649-028 Lisboa, Portugal; dfontinha@medicina.ulisboa.pt (D.F.); raquel.azevedo@medicina.ulisboa.pt (R.A.); mprudencio@medicina.ulisboa.pt (M.P.); 5Grupo de Química Inorgânica Medicinal e Reações Aplicadas, Instituto de Química, Universidade Federal do Rio Grande do Sul (UFRGS), Porto Alegre 91501-970, Brazil; wilmer.villarreal@ufrgs.br; 6Dipartimento di Scienze Biomediche, Chirurgiche e Odontoiatriche, Universitá degli Studi di Milano, 20133 Milan, Italy; nicoletta.basilico@unimi.it

**Keywords:** malaria, *Plasmodium*, hemozoin, heterobivalent, artemisinin, hybrids

## Abstract

A recently developed artemisinin-quinoline hybrid, named 163A, has been shown to display potent activity against the asexual blood stage of *Plasmodium*, the malaria parasite. In this study, we determined its in vitro cytotoxicity to mammalian cells, its potency to suppress *P. berghei* hepatic infection and to decrease the viability of *P. falciparum* gametocytes, in addition to determining whether the drug exhibits efficacy of a *P. berghei* infection in mice. This hybrid compound has a low level of cytotoxicity to mammalian cells and, conversely, a high level of selectivity. It is potent in the prevention of hepatic stage development as well as in killing gametocytes, denoting a potential blockage of malaria transmission. The hybrid presents a potent inhibitory activity for beta-hematin crystal formation, in which subsequent assays revealed that its endoperoxide component undergoes bioactivation by reductive reaction with ferrous heme towards the formation of heme-drug adducts; in parallel, the 7-chloroquinoline component has binding affinity for ferric hemin. Both structural components of the hybrid co-operate to enhance the inhibition of beta-hematin, and this bitopic ligand property is essential for arresting the growth of asexual blood parasites. We demonstrated the in vivo efficacy of the hybrid as an erythrocytic schizonticide agent in comparison to a chloroquine/artemisinin combination therapy. Collectively, the findings suggest that the bitopic property of the hybrid is highly operative on heme detoxification suppression, and this provides compelling evidence for explaining the action of the hybrid on the asexual blood stage. For sporozoite and gametocyte stages, the hybrid conserves the potency typically observed for endoperoxide drugs, and this is possibly achieved due to the redox chemistry of endoperoxide components with ferrous heme.

## 1. Introduction

Malaria, caused by the unicellular pathogen *Plasmodium*, still represents a major public health problem worldwide. In the last few years, the overall number of cases and estimated deaths due to malaria has remained constant rather than declined [1]. This is most likely due to a combination of factors, including interruption of public health and basic assistance, civil wars, resistance to multiple drug therapies, massive movement of refugees, and changes in climate conditions [2]. Yet, in 2019, 27 countries were able to eliminate malaria transmission, a remarkable number in comparison to only six countries in 2000. This outcome is most likely due to a solid implementation of public health programs to combat malaria using insecticide-treated mosquito nets and newer antimalarial medicines [1,2,3].

Many scientific studies spanning from basic parasitology to applied pharmacology have been performed to understand how antimalarial medicines work and how the parasites can escape treatment through drug resistance mechanisms [4,5]. Among them, the obligate parasite’s heme detoxification process has received considerable attention [6,7]. An example of masterly evolutionary adaptation by an intracellular parasitic pathogen, the parasite digests at least 80% of red blood cell hemoglobin as it grows inside the unfriendly environment of mature and/or immature human red blood cells, depending on the species. Hemoglobin digestion releases cytotoxic ferrous heme (iron II protoporphyrin, Fe^II^-PPIX), which is oxidized to ferric hematin (iron III protoporphyrin, Fe^III^-PPIX) and detoxified by crystallization to an inert form known as hemozoin (Hz, also known as malaria pigment) which is harmless to the parasite [6,7]. This detoxification process essentially converts a soluble iron-containing molecule into an insoluble β-hematin dimer, which then adds to growing Hz crystals [6,7]. As the parasite grows and destroys human red blood cells, lysed cells release Hz crystals into the bloodstream, where Hz is sequestered and deposited within the deep endothelial vasculature. Mounting evidence has shown that tissue inflammation that typically accompanies the disease is in part due to Hz deposition in the inflammatory infiltrate. This is associated with the acute stage of malaria, which may present severe manifestations [4,8].

*Plasmodium* is one of only three genera on the planet that produce Hz and depends on its formation as a mechanism of heme detoxification [6,7]. Hz appears as a unique target for antimalarial drug therapies, as the pharmacological blockage of this process is deleterious for parasite survival. This has been the main mode of action of antimalarial quinolines quinine, chloroquine, and amodiaquine, among other drugs used for treating patients, for many decades. These drugs slow the growth of the parasite by targeting multiple steps of the heme detoxification (Figure 1). The underlying mechanisms this action may include: targeting the soluble ferric hematin to result in an augmentation of free hematin levels, which is cytotoxic for parasites; targeting the soluble ferric hematin by forming even more cytotoxic drug–heme complexes; and finally adsording the drugs to the Hz crystal surfaces, blocking its growth. All these mechanisms involve drug binding to ferric hematin, which ultimately leads to an increase in the levels of the exchangeable heme species [9,10]. In contrast, antimalarial peroxides, such as artemisinins and ozonides, lack strong affinity for ferric hematin but undergo a mandatory process of activation that entails drug bioreduction through a redox reaction with intracellular ferrous heme, involving intermediate radical species and ultimately producing heme-drug adducts [11,12,13,14]. New compelling evidence has shown that the intermediate species can cause a radical-induced protein alkylation by leading to inactivation of proteins essential for the parasite survival, whereas the heme-drug adducts are cytotoxic to the parasites by suppressing the heme detoxification process [15].

In this context, the combination of antimalarial peroxides with antimalarial quinolines offers a unique possibility of spanning multiple ways to suppress heme detoxification and simultaneously act through an irreversible mechanism of alkylation of essential parasite proteins to ultimately achieve the antiplasmodial activity. Yet, the parasite may be able to escape this therapy through drug resistance mechanisms. To overcome this, different strategies have been considered, such as the addition of a third drug partner [16], prolonged therapy [17], improved pharmaceutical formulations [18], and chemically conjugation of drug partners [19,20]. The last strategy typically includes the molecular hybridization of antimalarial structures towards a single molecule of bitopic ligand property (i.e., bivalency in regard to pharmacophore property), which can offer a simpler pharmacokinetic profile than that of drug combinations while conserving the pleiotropic antimalarial effects of the drug combination [21,22].

We and others have demonstrated that various classes of antimalarial compounds can be hybridized with antimalarial peroxides such as artemisinins and ozonides to form hybrid-based drugs, resulting in potent activity due to summation of effects [13,23,24,25,26,27,28,29]. To identify the optimal position for molecular combination and the chemical linkers for molecular hybridization, we recently launched a campaign to design peptide-cleavable linkers, non-cleavable linkers, and molecular combinations by using refined synthetic methodologies [27,29]. This work resulted in the discovery of a potent antimalarial hybrid of artemisinin and 7-chloroquinoline, the hybrid-based drug named 163A [29] (Figure 2). The 163A hybrid presented superior potency for the asexual blood stage of *P. falciparum* in comparison with established single drugs. Furthermore, the potent in vitro antimalarial activity of 163A was translated into excellent efficacy against *P. berghei* in a mouse model, revealing higher efficacy than established single drugs [29]. Despite the progress achieved by this and other antimalarial hybrid-based drugs in recent years, little is known regarding the drug profiling towards stages of the *Plasmodium* life cycle other than the asexual blood stage. Another fundamental question relies on whether hybrid-based drugs, theoretically designed as heterobivalent molecules, can indeed exhibit a bitopic ligand property on heme detoxification process by simultaneously displaying affinity for ferric hematin, as classically observed for quinolines, as well as undergo redox chemistry reactivity with ferrous heme, as typically observed for endoperoxide antimalarials like artemisinin. Here, we aimed to study the antiplasmodial therapeutic breadth and drug activity profile for this hybrid-based drug and to assess its potency and efficacy compared to those of established single drugs or drug combinations.

## 2. Results

### 2.1. Toxicity for Host Cells and Selectivity

The 163A hybrid has been previously shown to inhibit *P. falciparum* parasite growth by both CQ-sensitive and CQ-resistant strains of the asexual blood stage [29]. We sought to assess the cytotoxicity of 163A by using both J774 (a non-cancerous macrophage cell line) and HepG2 (hepatic cells as potential host for sporozoites) lineages under the same conditions used to determine antimalarial potency. Toxicity of 163A was assayed in parallel to established single antimalarial drugs artemisinin and chloroquine as representative parental drug components of 163A, and doxorubicin was used as a standard cytotoxic drug. These experiments were subsequently used to estimate an in vitro selectivity index (S.I.), which measures the 163A concentration resulting in 50% cytotoxicity (CC_50_) over the concentration needed for 50% parasite growth inhibition (IC_50_) (Figure 3).

Like the parental drugs, 163A hybrid showed cytotoxicity for non-infected cells in the micromolar range, being at least 15 fold less cytotoxic than anticancer agent doxorubicin. Next, we determined the S.I. for the individual components and observed that in HepG2 cells, the 163A hybrid is 3 to 4 fold more selective than parental drugs. For J774 cells, 163A hybrid exhibited twice more cytotoxicity than chloroquine, whereas artemisinin is devoid of cytotoxicity in concentrations up 200 µM. Despite superior cytotoxicity than parental drugs, the 163A hybrid is still several fold less cytotoxic than anticancer agent doxorubicin. For selectivity index calculated for J774 cells, the 163A hybrid and its parental drugs presented a similar selectivity. The molecular hybridization of artemisinin and 7-chloroquinoline towards the 163A hybrid did not negatively affect the toxicity profile when compared to parental drugs. A previous study has suggested an augmentation of the toxicity profile of antimalarial hybrids in comparison to the parental drugs, possibly related to the chemical nature of the linker [26], however, we can infer that the chemical linker, employed on 163A structure, has cytocompatibility to a cellular environment.

### 2.2. Inhibition on Hepatic and Sexual Stages

In addition to its activity against the parasite’s asexual blood stages, it is desirable for a novel antimalarial agent to also demonstrate activity against the hepatic stage of infection, which is obligatory for the onset of blood infection and, depending on the parasite species, can be involved in disease relapses (as is the case of *P. vivax* infection) [30,31]. Additionally, activity against the sexual blood stages of *Plasmodium* is desirable for a novel antimalarial, as these stages are involved in parasite transmission to the invertebrate host [32,33]. To address this, we used an in vitro model of Huh-7 hepatic cells infected with *P. berghei* sporozoites to determine the activity of 163A against hepatic infection [34]. For the sexual blood stage of *P. falciparum* 3D7 strain, a standard protocol for gametocytogenesis was employed [35], and gametocyte viability was measured by bioluminescence of luciferase (Figure 4).

Our results show that 163A inhibits hepatic infection but has no impact on the viability of the host hepatic cells. In terms of potency, it is 10-fold more potent to inhibit *Plasmodium* liver schizont development than the standard drug in the therapy (primaquine) and it is as potent as artesunate. In marked contrast, the parental drug artemisinin has low inhibitory activity for liver schizonts [36,37] and dihydroartemisinin has variable inhibitory activity depending on *Plasmodium* species and host, in part due to its low stability [38]. In addition, 4-aminoquinolines such as chloroquine present low potency in the hepatic stage of *Plasmodium* infection [39], suggesting that the observed inhibitory activity of 163A against this phase of the parasite’s life cycle can be attributed to the molecule’s endoperoxide compartment.

Next, an assessment of the effect of 163A on the viability of gametocytes was performed for young (stages I to III) and mature gametocytes (stage V). For young gametocytes, results showed that this was strongly inhibited by both dihydroartemisinin and the 163A hybrid, with IC_50_ values in low nanomolar range and similar to observed for IC_50_ values against the asexual blood stage. In contrast, both compounds presented potency in micromolar range for mature gametocytes. For both young and mature gametocytes, the 163A hybrid was 2- to 3-fold less potent than dihydroartemisinin, whereas chloroquine is at least 20-fold less potent than dihydroartemisinin for arresting viability in mature gametocytes. Based on this, it is possible to deduce that the 163A hybrid preserves the gametocyte killing property typically observed for antimalarial endoperoxides.

### 2.3. Dual Role on the Heme Detoxification Process

The bitopic ligand property of the 163A hybrid is related to the feasibility of endoperoxide to target ferrous heme and the 7-chloroquinoline to target ferric hematin, possibly generating a single heterobivalent molecule able to display a dual role in the heme detoxification process. The 163A hybrid has been shown to inhibit β-hematin formation and to suppress the in vivo parasite heme detoxification [29]; however, this heterobivalency property was not previously examined in terms of 163A targeting ferrous heme and its underlying redox chemistry reactivity. To examine this heterobivalency property (Figure 5), we employed a recent methodology that allows the simultaneous examination of β-hematin inhibitory activity (BHIA) under an oxidizing condition (ferric hematin) and under a reducing condition (ferrous heme) [40], in addition to determining the association constant of drugs for ferric hematin (log *K*) [41].

The 163A hybrid greatly inhibited β-hematin formation assayed using the conventional ferric hematin by displaying a potency twice as high as chloroquine, whereas artemisinin lacked inhibitory activity. The 163A hybrid was as potent as chloroquine in inhibiting β-hematin formation under ferrous heme and it was only slightly less potent than artemisinin, which is a strong inhibitor assayed in the presence of ferrous heme. The inhibitory activity on the β-hematin formation was higher when assayed for ferrous heme than ferric hematin, which is also a behavior observed for 7-chloroquinoline [40]. Other hybrid-based drugs have displayed potency equal or superior to parental drug and antimalarial quinolines of reference, such as amodiaquine and chloroquine [13,26]. Consistent with this notion and based on a model of structural determinants for 7-chloroquinolines inhibiting β-hematin formation [42], a plausible reason for increased inhibitory potency of hybrid for β-hematin crystal formation is a combination of binding affinity of 7-chloroquinoline component for ferric Fe-PPIX and the chemical linker playing a role as a side chain by assisting further binding property.

To understand the reason for the 163A hybrid to inhibit β-hematin formation in an oxidizing condition, measurement of the association constant for ferric hematin (log *K*) was determined. Titration of the 163A hybrid to ferric hemin revealed that it has binding affinity by suggesting a hematin–drug complexation, with a log *K* value lower than observed for chloroquine, whereas artemisinin did not exhibit binding property. This result indicates that binding of 163A to ferric hemin is weaker than to chloroquine. Literature has shown the aminoalkyl side chain of chloroquine is essential for its binding to Fe-PPIX [42] and based on this, a formal possibility is that the chemical linker of the 163A hybrid may not reproduce an optimal binding affinity as observed for chloroquine. Furthermore, the speciation of hematin–drug complexation observed in titration experiments may not be the same as observed for β-hematin assays, justifying the increased potency of hybrid for inhibiting β-hematin formation than chloroquine.

### 2.4. Parasitemia Profile in P. berghei-Infected Mice

Typically, a single drug therapy requires a much higher dose for curing established infection (Thompson test) than suppressing onset infection (Peters test). However, 163A has previously demonstrated success as an efficacious antimalarial by observing that the same 163A dosage for suppressing parasitemia also presented curative potential in an established infection [29]. As a limitation, 163A treatment was previously studied by subcutaneous administration, which is an optimal route of high absorption for lipophilic drugs such as artemisinin, but here, we wanted to determine in a more stringent way whether 163A could result in a decline in parasitemia and could cure mice when administered by intraperitoneal injection, a route of drug administration that is typically employed for severe malaria patients. In *P. berghei*-infected Swiss mice with a patent parasitemia > 3.0%, we evaluated in a 3-day dose regime at 140 μmol/kg of 163A or a fixed-dose drug combination (chloroquine plus artemisinin). Twenty-four hours after cession of treatment (day 9 post-infection), parasitemia was measured in different intervals, in addition to a follow up of animal survival up to 40 days post-infection.

As observed in Figure 6, 163A was efficacious at inhibiting parasitemia in comparison to the untreated group, with a reduction larger than 89% at the peak of parasitemia. In the group receiving the drug combination, parasitemia reduction was > 99% in comparison to the untreated group, which is consistent with the notion that this drug combination has summation of efficacies on reducing parasitemia in *P. berghei*-infected mice [43]. Importantly, we found that 24 h after cession of treatment, 163A or drug combination treatment was able to decrease parasitemia, a feature typically observed for artemisinin but not for chloroquine. The period of maximum parasitemia reduction for 163A was 96 h posttreatment. Recrudescent parasite was monitored daily and observed at day 14 for 163A therapy and at day 19 for drug combination. The median survival time was twice as high for the drug combination in comparison to the untreated group or the group receiving 163A treatment.

Consistent with previous reports in the Thompson malaria model [43,44], therapy using a single antimalarial drug given once-a-day does not cure mice. Exceptions are the therapies of high-dose regimens, multiple daily doses, drug combinations, and specific route of administration (such as the subcutaneous administration for highly lipophilic drugs). Based on this, we inferred that in comparison to subcutaneous administration, intraperitoneal administration of 163A at the same dose regime provides a sharp decline of parasitemia; however, with intraperitoneal administration this drug dosage is not enough to resolve infection and cure mice. In addition, we inferred that the non-superior efficacy of 163A observed in comparison to drug combinations is probably because the 7-chloroquinoline component from the 163A structure is a less potent antimalarial than chloroquine employed in the drug combination [29].

## 3. Discussion

The presented findings of potency and selectivity of hybrid 163A as an antimalarial agent for the multiple stages of *Plasmodium* life cycle, illustrated in Figure 7, argues in favor of drug development of hybrid-based drugs in comparison to single antimalarial drugs. For asexual blood stage of *P. falciparum*, hybrid 163A presents superior potency and higher selectivity indexes in comparison to single antimalarial drugs. We interpreted that both components, 7-chloroquinoline and endoperoxide, can suppress the heme detoxification process, which is a biochemical process highly active in the asexual blood stage. For sexual and hepatic stages, hybrid 163A presents potency and selectivity indexes in comparison to endoperoxide drugs artesunate/dihydroartemisinin, whereas the quinoline component is typically devoid of antiparasitic activity for these stages.

For the asexual blood stage, the heme detoxification process is an important drug target. It has been well known for decades that 4-aminoquinolines such as chloroquine and amodiaquine strongly suppress this process in multiple ways and target soluble ferric hematin and adsorb on surfaces of hemozoin crystals [6,7]. However, only in recent years has it become clear that antimalarial endoperoxides, such as artemisinin and ozonides, can suppress the heme detoxification process by a redox chemical process involving ferrous heme towards the formation of heme-drug adducts. This not only leads to a depletion of cytosolic heme levels and alters iron redox homeostasis but also produces heme-drug adducts that are endogenous antiparasitics [16]. Based on this, a more compelling hypothesis is that the enhanced antimalarial profile, i.e., the summation of effects by additivity or a synergism phenomenon, for drug combination between 4-aminoquinolines and endoperoxides is in part due to pleiotropic effects on the heme detoxification. The summation of effects has important consequences for antimalarial activity at the asexual blood stage. Consistent with this notion, the 163A hybrid presents superior potency, higher selectivity indexes, and stronger efficacy in comparison to single antimalarial drugs. Our experimental approach using multiples assays on soluble Fe-PPIX and β-hematin crystal formation, which is a compelling model to study the *Plasmodium* heme detoxification, argues that both components, 7-chloroquinoline and endoperoxide, can simultaneously act to suppress the heme detoxification process.

We observed that 163A has a potent antiparasitic activity against the liver stage. Typically, 8-aminoquinolines such as primaquine and tafenoquine have antiparasitic activity for the liver stage [45] whereas 4-aminoquinolines are devoid ofantiparasitic activity for this stage. For antimalarial endoperoxides, artemisinin lacks inhibitory activity for the liver stage when assayed up to 10 µM, whereas artesunate and artemisone inhibit liver schizonts [37]. Based on this, we can infer that the 163A hybrid activity is mostly due to the presence of the endoperoxide compartment, as artesunate is active whereas chloroquine is not. It was previously observed that an artemisinin/primaquine hybrid-based drug containing a cleavable-linker displayed enhanced potency and efficacy for *Plasmodium* liver stage in comparison to an analogous hybrid containing a non-cleavable-linker or to single parental drugs [36], which is in line with the observations disclosed here for hybrid 163A. Our data indicate that the hybrid can preserve the activity against the liver stage of antimalarial endoperoxides, which is a remarkable feature if considering not all antimalarial endoperoxides can inhibit liver stage parasites. Moreover, considering the importance of this stage for controlling infection and relapse, further investigation into the potency and efficacy of the 163A hybrid is required to address whether it fully eliminates liver stage parasites or solely delays their development.

We further observed that the 163A hybrid has a strong antiparasitic activity for young gametocytes, with potency comparable to dihydroartemisinin, the reference drug in the assay. The potency of these compounds for young gametocytes is very close to the asexual blood stage, denoting a potential for blocking parasite development towards mature gametocytes and possibly reducing gametocyte carriage in patients. Our findings are consistent with a previously described hybrid-based drug composed of 1,2,4-trioxane and quinoline that exhibited stronger activity for gametocytes than the quinoline component and the same potency as its endoperoxide component [23].

For less metabolically active sexual parasites in stage V, endoperoxide compounds as well as other clinically relevant antimalarials typically require a higher drug concentration to kill them [35,46,47]; an exception is the experimental drug methylene blue. Although certainly the 163A hybrid and dihydroartemisinin have inferior efficiency against late stage, of note, both drugs presented IC_50_ values ranging 2–5 µM, whereas the maximum concentration of dihydroartemisinin in patient serum is approximately 4 µM [48], a concentration closer to IC_50_ values necessary for killing mature gametocytes. Furthermore, we have inferred that, as in the liver stage, the activity against gametocytes is greatly influenced by the endoperoxide component, as chloroquine is devoid of gametocytocidal activity. To fully assess 163A action on late stage, further experiments using exflagellation are necessary, but to date the potency of 163A against immature gametocytes denotes a potential for blocking malaria transmission to the invertebrate host.

Regarding the mode of action, the heme detoxification process is one of the parasite targets for 163A against asexual blood stages, followed by a 163A bioreductive activation through heme towards the process of radical-induced protein alkylation [29]. For the mode of action against the liver and sexual stages, this remains less clear. Gametocytes are nonreplicating and lack substantial hemozoin synthesis as compared to the asexual blood stage; in addition, in the liver stage, the hepatic cells have less abundant cytosolic heme than red blood cells, which is necessary for bioreductive activation of artemisinin by heme. Despite differences in parasite metabolism and heme abundance within the *Plasmodium* life cycle, a plausible explanation is that antimalarial endoperoxides, including hybrid 163A, may arrest parasite growth by the same mechanism of drug bioreductive activation *via* heme and alkylation of proteins and heme. Apart from this mechanism, another plausible explanation for the preserved or even superior potency of hybrid-based drugs in comparison to single quinoline drugs is the enhanced lipophilicity for the hybrid 163A in comparison to parental drugs, possibly enhancing drug permeability and accumulation. In line with this, a recent work has indicated that enhanced lipophilicity is an important aspect for explaining superior antimalarial activity of homobivalent drugs based on artemisinin dimers in comparison to monomers [49].

Previously, we demonstrated the superior efficacy of 163A therapy in comparison to single antimalarial drugs [29]. Here, we extended the efficacy profile by testing 163A therapy using intraperitoneal administration, performed in parallel to a group receiving a fixed-dose of drug combination. Both therapies produced a sharp decline in parasitemia in comparison to untreated infected mice, and the drug combination therapy was able to prolong animal survival. The 163A therapy demonstrated non-superior efficacy to drug combination and the reason for this could be that the in vivo metabolization of ester chemical linker of 163A hybrid to yield the 7-chloroquinoline component (compound 2, Figure 2) [29]. At least in vitro, this 7-chloroquinoline component is a less potent antimalarial agent than the quinoline component of the drug combination (chloroquine). The non-superior efficacy of hybrid-based therapy versus drug combination has also been observed by others [23,24], and this profile does not undermine the antimalarial potential of this class of drugs. The initial concept of hybrid-based therapy was to replace fixed-dose drug combinations [50]; however, a new potential use is to replace one of the components of drug combinations with a hybrid-based drug.

## 4. Materials and Methods

### 4.1. General Materials

Chloroquine diphosphate, artemisinin, and hemin chloride (BioXtra) were obtained from Sigma-Aldrich (St. Louis, MO, USA). Primaquine phosphate was manufactured by FarManguinhos (Rio de Janeiro, Brazil). May-Grunwald-Giemsa stain was obtained from the Panotico (Laborclin, Pinhais, Brazil). Tribromoethanol was supplied from Sigma-Aldrich (St. Louis, MO, USA) and dissolved in 2-propanol:water 20% (*v*/*v*). All other general laboratory chemicals and solvents were of analytical or HPLC grade. Experimental drug 163A was synthesized and purified as previously described [29].

### 4.2. Cytotoxicity Assay in Mammalian Cells

Compounds were tested for in vitro cytotoxicity against two mammalian cell lineages, J774 (murine macrophages) and HepG2 (human hepatocellular carcinoma), and cellular viability was determined by bioluminescence using CellTiter-Glo kit (Promega, Madison, USA). Lineages were maintained in RPMI-1640 (hepG2) or DMEM (J774) containing 10% fetal bovine serum and supplemented with L-glutamine, vitamins, and amino acids in 75-cm^2^ flasks at 37 °C, with the medium changed twice weekly. Cells from 60 to 80% confluent cultures were trypsinized, washed in complete medium, and plated at 4 × 10^4^ cells per well in 100 µL of complete medium in 96-well flat-bottom white plates for 24 h at 37 °C prior to the addition of the compounds. Triplicate aliquots of compounds and the reference drugs (stock solution in DMSO) covering 7 different concentrations (200–3.125 µM) at 2-fold dilutions were added to the wells, and plates were incubated for 72 h more. Controls included compound-free wells with DMSO (vehicle) as positive controls and cell-free wells with medium only, which were used for background subtraction. Following incubation for 72 h at 37 °C, plates were maintained at room temperature, the culture medium was removed, and a 100 µL volume of CellTiter-Glo kit was added to each well. The bioluminescence was measured using a microplate reader Filtermax™ F3 & F5 Multi-Mode instrument (Molecular Devices, San Jose, USA) and Softmax software. CC_50_ data were obtained from at least two independent experiments for each cell line and analyzed using GraphPad Prism Version 5.01. The selectivity index (S.I.) was calculated using the CC_50_s from mammalian cells divided by the IC_50_ obtained against the asexual blood stage *P. falciparum* 3D7.

### 4.3. P. berghei Liver Stage

Human hepatoma cell line (Huh-7) cells were cultured in 1640 RPMI medium supplemented with 10% *v*/*v* fetal bovine serum, 1% *v*/*v* nonessential amino acids, 1% *v*/*v* penicillin/streptomycin, 1% *v*/*v* glutamine, and 10 mM HEPES, pH 7, and maintained at 37 °C with 5% CO_2_. Huh-7 cells at 1.0 × 10^4^ per well were seeded in 96-well plates the day before drug treatment and infection. The medium was replaced by medium containing the appropriate concentration of each compound approximately 1 h prior to infection with sporozoites of firefly luciferase-expressing *P. berghei* line freshly obtained through disruption of salivary glands of infected female *Anopheles stephensi* mosquitoes. An amount of the DMSO solvent equivalent to that present in the highest compound concentration was used as a control. Sporozoite addition was followed by centrifugation at 1700× *g* for 5 min. Parasite infection load was measured 48 h after infection by a bioluminescence assay (Biotium, Hayward, CA, USA). The effect of the compounds on the viability of Huh-7 cells was assessed by the AlamarBlue assay (Life) using the manufacturer’s protocol.

### 4.4. P. falciparum Gametocytes

Drugs were serially diluted in a 96-well flat bottom plate (concentration range 29.0–0.22 μM) in 100 μL per well, and each drug was tested in duplicate, in seven different concentrations. A volume of 100 μL of *P. falciparum* gametocytes from the luminescent strain 3D7elo1-pfs16-CBG99 (stages I to III or stage V) at 0.5–1% parasitemia and 2% hematocrit were dispensed. For mature gametocytes, a culture containing more than 90% of stage V gametocytes was employed. Methylene blue was used as positive control. Luciferase activity was used for measuring gametocyte viability. Plates were incubated for 72 h at 37 °C under 1% O_2_, 5% CO_2_, and 94% N_2_ atmosphere. A volume of 100 µL of culture medium was removed from each well to increase hematocrit, and 70 µL of resuspended culture were transferred to a black 96-well plate. A volume of 70 µL of D-luciferin (1 mM in citrate buffer 0.1 M, pH 5.5) was added. Luminescence measurements were performed after 10 min with 500 ms integration time using a microplate reader Sinergy 4 (Biotek). The IC_50_ was extrapolated from the non-linear regression analysis of the concentration–response curve. The percentage of gametocytes viability was calculated as 100 × [(OD treated sample − OD blank)/(OD untreated sample − µc-blank)] where “blank” is the sample treated with 500 nM of methylene blue, which completely kills gametocytes.

### 4.5. β-Hematin Inhibition Activity (BHIA)

A 10 μL sample of hemin chloride at 4 mM dissolved in 80% DMSO was aliquoted to a U bottom 96-well microplate. A 10 μL sample of 0.1 M NaOH (O-BHIA) or a 4 mM reduced glutathione in PBS at pH 7.0 (R-BHIA) was then added and plates were sealed and incubated at 37 °C for 2 h. A 10 μL sample of drug dissolved in DMSO were added and after incubation for 2 h, 180 μL of acetate buffer (2.0 M, pH 5.2) and 10 μL of nonionic detergent IGEPAL-CA-630 solution in methanol (10 mg/mL) were added, plates were sealed and incubated at 37 °C for 18 h. Each drug was tested in a range from 1000 to 17 μM at final concentration. The reaction was stopped by the addition of 50 μL of a sodium dodecyl sulfate (5.0% *v*/*v* in 0.1 M bicarbonate buffer at pH 9.0). Plates were centrifuged at 3700 rpm for 5 min., and a 100 μL of supernatant was placed on a separate plate containing 100 μL/well of a pyridine solution (20% *v*/*v* in HEPES 1 M) according to a previous method [51]. Absorbance was quantified at 405 nm using a SpectraMax 190 microplate reader (Molecular Devices, Sunnyvale, CA). The percentage of inhibition was compared to untreated and unreacted controls (no acetate buffer), and IC_50_ values were estimated by using a non-linear regression curve calculated in GraphPad Prism. At least two independent experiments were performed, each concentration in quadruplicate.

### 4.6. Determination of the Association Constant to Ferriprotoporphyrin IX

The association constant of compounds to ferriprotoporphyrin IX (hemin chloride) was measured essentially as previously described [41]. Titration of a 2 mL solution (7.5 µM of ferriprotoporphyrin IX in 40% of DMSO, apparent pH 7.5) in presence of compound (500 µM in 40% of DMSO, pH 7.5) was performed by UV absorbance at λ = 402 nm using Hewlett Packard spectrophotometer, diode array model 8452 (Santa Clara, CA, USA). The volume of each titration was 5 µL, and the relative molar ratio varied from 0 to 10 with regard to [Fe^III^-PPIX]. Spectra were recorded about 1 min after each addition. The absorption of all compounds was subtracted by adding the same amounts to the blank (40% of DMSO, pH 7.5). Fitting model with a 1:1 drug:association was used by the equation described by Egan [41]:A = A_0_ + A *K*_∞_ [C]
1 + *K* [C]
where A_0_ is the absorbance of hemin before addition of complex or free chloroquine, A_∞_ is the absorbance for the drug-hemin complex at saturation, A is the absorbance at each point of the titration, and *K* is the conditional association constant. Three independent experiments were performed.

### 4.7. Mice

Male Swiss mice (4 to 8 weeks old) for experimental study were obtained from the Animal Resource Facility at Instituto Gonçalo Moniz (Salvador, Brazil). Male or female BALB/c mice (7 to 8 weeks old) were used for weekly passage of malaria parasites. Animals were housed under a 12-h light/dark cycle with free access to sterilized food pellets and sterilized water. These studies were approved by the Instituto Gonçalo Moniz Animal Ethics Committee (protocol 014/2018).

### 4.8. Parasites

Drug-sensitive *P. berghei* parasites expressing green fluorescent protein (GFP) [52] were maintained by continuous weekly blood passage in Swiss mice. Mouse was euthanized by tribromoethanol (300 mg/kg animal weight), blood was collected by cardiac puncture, and a standard inoculum of 10^7^ parasitized erythrocytes per 200 µL was prepared by dilution in 0.9% saline and administered by intraperitoneal (i.p.) injection to experimental mice. Parasite enumeration in infected mice was determined in the peripheral blood collected from the tail veins. For determination of parasitemia in passage mice, blood smears were mounted in slides and stained by May-Grunwald-Giemsa. For experimental study, parasite enumeration was performed by flow cytometry using GFP signal and co-staining with 5.0 nM of Mitotracker deep red FM (Life Invitrogen) for 15 min. of incubation.

### 4.9. P. berghei-Infected Mice (Thompson Test)

Male Swiss mice were inoculated with a 2 × 10^6^ parasitized erythrocytes per 200 μL by i.p. injection. After infection reaches a parasitemia between 3 to 5% (determined by flow cytometry, BD LSRFortessa™ Cell Analyzer, Piscataway, NJ, USA), mice were randomly divided in *n* = 6/group and treated once-a-day per 100 μL by i.p. injection of respective drug or vehicle for three consecutive days. Drugs were solubilized in DMSO and diluted in a solution of Kolliphor (Cremophor, 4%), Polysorbate 80 (5%), Sorbitol (5%), glucose (5%), and Tween 20 (5%) in 0.9% saline to a final concentration of 10% (*v*/*v*) of DMSO. Experimental compound 163A was administered at 140 µmol/kg of animal weight (82.5 mg/kg), a fixed-dose of drug combination (chloroquine plus artemisinin) was administered at 140 µmol/kg of animal weight (using 70 µmol/kg each drug), and the untreated group received vehicle only. Twenty-four hours after the last drug administration, parasitemia was determined in regular intervals for 5 days. Animal survival was monitored twice a day for 40 days after infection. Parasitemia reduction was determined in comparison to the vehicle drug. One single experiment was performed.

### 4.10. Statistical Analysis

Data are presented as median ± standard error of the mean (S.E.M.) or mean ± standard deviation (S.D.). Statistical analysis was performed using the Prism version 5.01 software (GraphPad Software, La Jolla, CA, USA). Statistical significance was assessed by performing the one-way analysis of variance (ANOVA) followed by post-test for multiple comparisons as indicated in each figure. Differences with *p* values < 0.05 were considered significant.

## 5. Conclusions

We found that 163A has low cytotoxicity for host cells and has a broad inhibition profile spanning the *Plasmodium* life cycle, including hepatic stage and gametocytes, which indicates a potential of this drug to block malaria transmission. The broad spectrum of antimalarial activity is achieved in part because the hybrid has pleiotropic effects on the heme detoxification, targeting both ferrous heme and ferric hematin. The efficacy of the hybrid is strong, superior to single antimalarial drugs, such as quinolines or artemisinins, and it is only reproduced when heterobivalent structures or a combination of drugs are employed. These findings support the continuation of drug development towards a single molecule with heterobivalency properties into antiparasitics for combating *Plasmodium* infection. The safety and potential of antimalarial profile of hybrid 163A merits further investigation of pharmacokinetics and efficacy in drug combination regimens.

## Figures and Tables

**Figure 1 pharmaceuticals-14-01129-f001:**
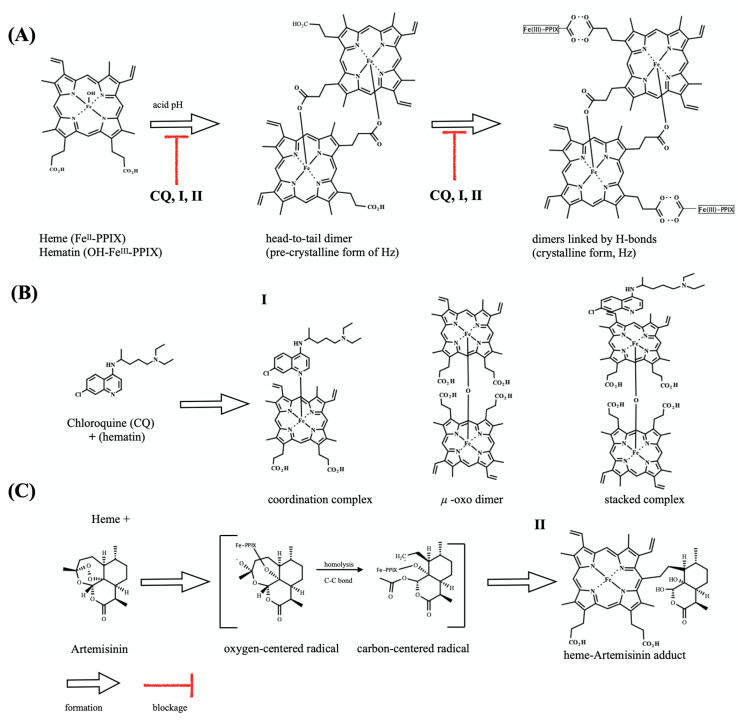
Overview of the heme detoxification process and drug blockage. (**A**) shows the dimerization of hematin through the reciprocal coordination of iron and propionate moieties and subsequent formation of hydrogen-bonds between dimers to form hemozoin (Hz) crystals. (**B**) shows the structure of chloroquine (CQ) and its speciation in the presence of hematin. Formation of hematin species μ-oxo-dimer is induced under CQ exposure and does not form Hz crystals. (**C**) shows the structure of antimalarial endoperoxide artemisinin, and its bioreductive activation by heme, leading to radical species responsible for alkylation of proteins as well as of Fe-PPIX. After alkylation, there occurs formation of a heme-artemisinin adduct. Unless specified, iron is in its ferric state.

**Figure 2 pharmaceuticals-14-01129-f002:**
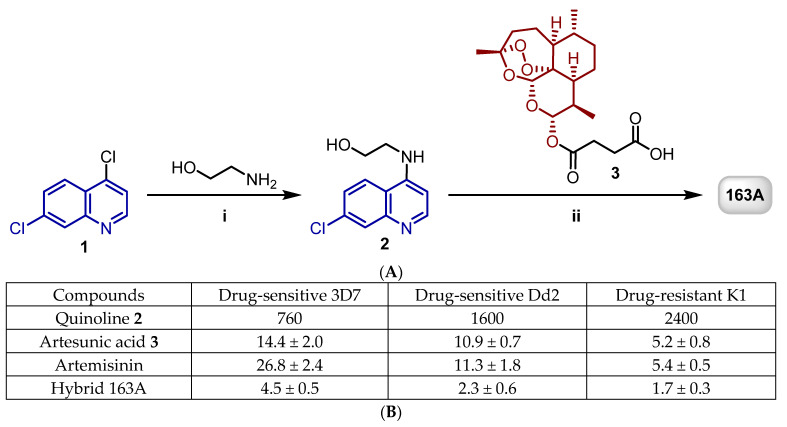
(**A**) shows the synthesis of artemisinin and 7-chloroquinoline hybrid 163A. Reagents and conditions: (i) Et_3_N, 120 °C, 2 h; (ii) DCC, DMAP, CH_2_Cl_2_, 0 °C to rt, 18 h, N_2_. (**B**) shows a table summarizing the IC_50_ values in nM against the asexual blood stage of *P. falciparum.* Adapted with permission from ref. [29]. Copyright 2019 John Wiley & Sons, Inc.

**Figure 3 pharmaceuticals-14-01129-f003:**
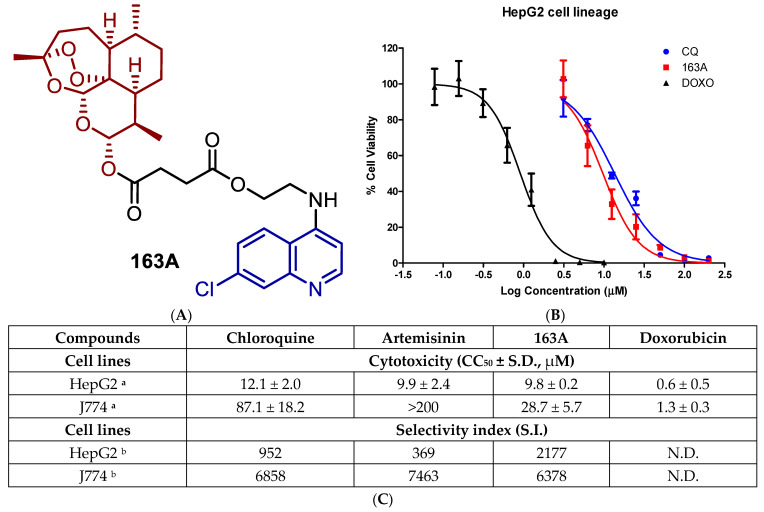
(**A**) Molecular structure of hybrid-based drug 163A highlighting the main structural components. (**B**) Uninfected HepG2 cells were cultured in the presence of the compound (163A, parental or reference drugs) for 72 h. Cell viability was detected by the ATP activity in the cell lysates by CellTiter-Glo. Data were normalized to the signal from untreated cells, which was set at 100% cell viability, and data were fit to a logistic curve by Prism version 5.01. (**C**) shows a Table with a summary of cytotoxic concentration for 50% (CC_50_) and selectivity index values. Footnotes for table: ^a^ Values are median ± S.D. of at least three independent experiments, each concentration tested in triplicate. ^b^ Selectivity index determined as CC_50_/IC_50_; values of IC_50_ performed for asexual blood stage of 3D7 strain of *P. falciparum* and determined after 72 h incubation time. CQ = chloroquine diphosphate; DOXO = doxorubicin. S.D. = standard deviation; N.D. = not determined.

**Figure 4 pharmaceuticals-14-01129-f004:**
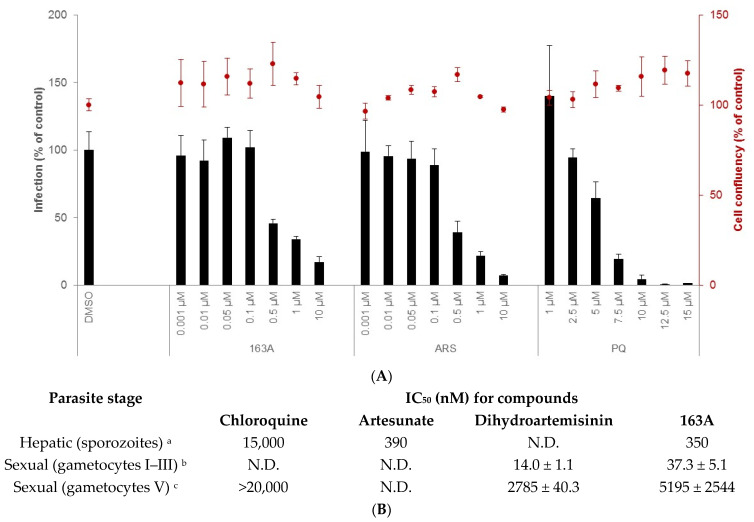
Characterization of 163A in vitro activity against hepatic sporozoites and sexual blood stage. (**A**) shows the % of infection (left y axis) and host cell confluency (right y axis) of Huh-7 cells infected by *P. berghei* sporozoites. (**B**) shows a table with a summary of inhibitory concentration for 50% (IC_50_) for multiple stages of the *Plasmodium* life cycle. Experimental details (footnotes in the table): ^a^ Activity was determined after 48 h of drug incubation. Two independent experiments were performed; values are mean of one experiment, each concentration tested in triplicate. Reference drug is primaquine (IC_50_ = 4700 nM). ^b^ Assayed against gametocytes at stages I to III of 3D7elo1-pfs16-CBG99 strain of *P. falciparum* and activity was determined after 72 h of drug incubation. ^c^ Assayed against gametocytes at V stage and the reference drug is Methylene blue (IC_50_ = 60 ± 10 nM). ^b,c^ Values are mean ± S.D. of three experiments, each concentration tested in duplicate. ART = artesunate; PQ = primaquine. N.D. = not determined.

**Figure 5 pharmaceuticals-14-01129-f005:**
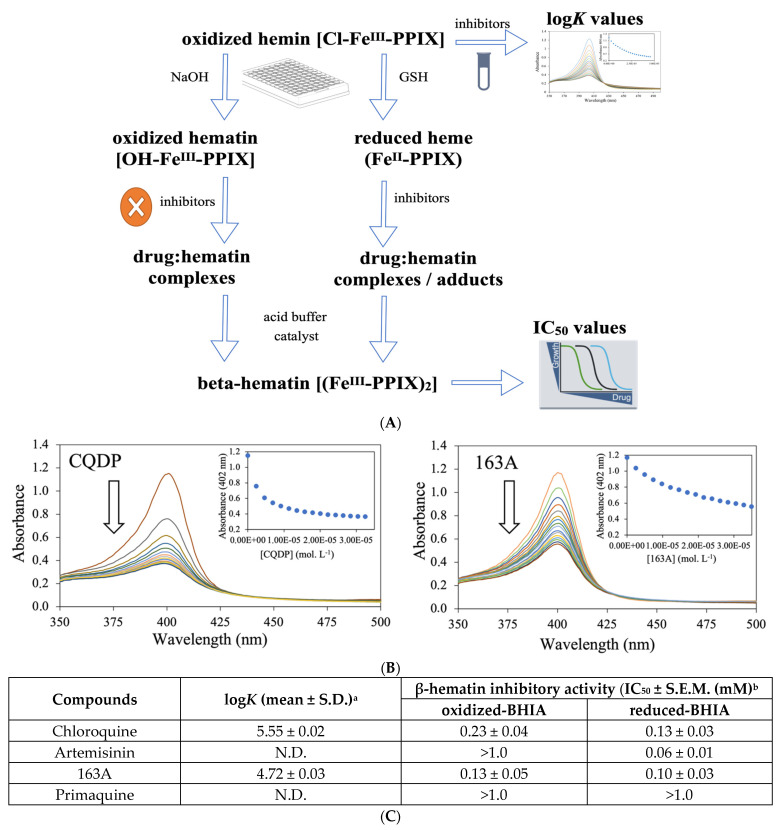
Dual role of 163A on the *Plasmodium* heme detoxification. (**A**) shows the experimental assays of β-hematin inhibitory activity (BHIA) for oxidized-BHIA and reduced-BHIA and subsequent assays. (**B**) shows titration of hemin to increasing concentration of drugs. Spectrophotometric characterization of hemin–drug complexes upon increasing drug concentration is indicated by the arrow. (**C**) is a table summarizing the association constant (log*K*) values of compounds for ferric hemin and inhibitory concentration for 50% (IC_50_) values of compounds for oxidized-BHIA and reduced-BHIA. Experimental details for table: ^a^ Log*K* values were calculated from three independent experiments. ^b^ IC_50_ were calculated from three independent experiments after 18 h of drug incubation. S.E.M. = standard error of the mean values. CQDP = chloroquine diphosphate; GSH = glutathione, reduced. N.D. = not determined due to lack of binding.

**Figure 6 pharmaceuticals-14-01129-f006:**
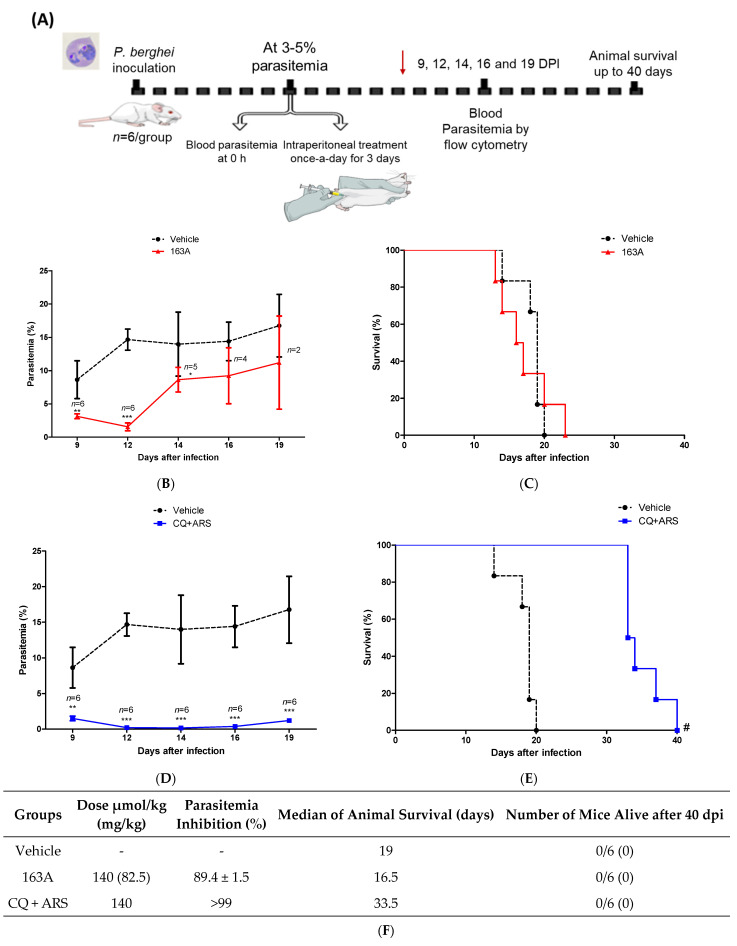
Efficacy of 163A in an established *Plasmodium* infection in mice. (**A**) Experimental design of Thompson therapy test. (**B**,**D**) Parasitemia profile in *P. berghei*-infected Swiss mice following therapy once-a-day by intraperitoneal administration of 163A or a fixed-dose drug combination (CQ + ARS). (**C**,**E**) Animal survival of *P. berghei*-infected Swiss mice following therapy. (**F**). Summary of parasite reduction and median of animal survival. Values are from one single experiment, using *n* = 6/group. Parasitemia inhibition was determined in comparison to vehicle, values are median ± S.D. Error bars indicate S.D. * *p* < 0.01, ** *p* < 0.001, *** *p* < 0.0001 (One-way ANOVA) versus vehicle; ^#^ *p* = 0.0007 (Log-rank and Mantel-Cox test) for CQ + ARS versus vehicle. DPI = days post-infection. Red arrow in panel A indicates the last day of drug administration.

**Figure 7 pharmaceuticals-14-01129-f007:**
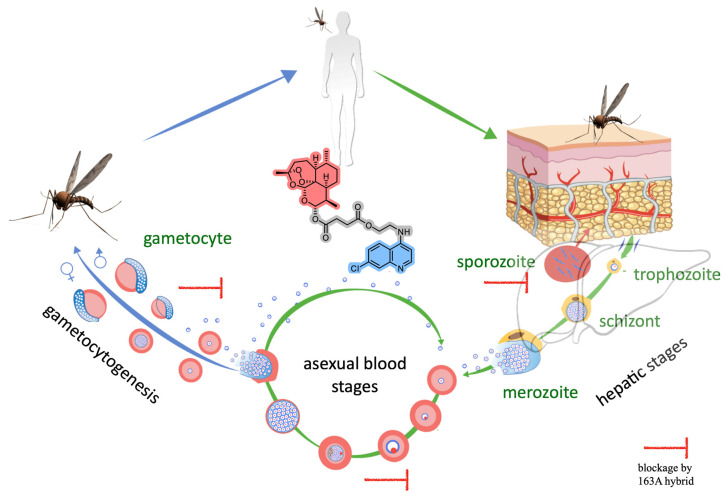
Overview of *Plasmodium* life cycle, denoting the three stages where the antimalarial activity of the 163A hybrid blocks the parasite growth and viability.

## Data Availability

Data is contained within the article.

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
