# Peer review of "Studies of Potency and Efficacy of an Optimized Artemisinin-Quinoline Hybrid against Multiple Stages of the Plasmodium Life Cycle"

_pharmaceuticals, 2021, doi:10.3390/ph14111129_

Round 1
Reviewer 1 Report
The manuscript presents the studies of potency and efficacy of an optimized dihydroartemisininquinoline hybrid against multiple stages of the Plasmodium life cycle.
In my opinion the content of this report is sufficient to be published in Pharmaceutical journal.
Author Response
Author's response: First of all, we are thankful for the reviewer's work in revising our manuscript. This question is very pertinent. At this stage, we could not assess the drug stability of 163A hybrid upon exposure to microsomal metabolism. However, we have now addressed this possibility in the latest manuscript version.
Reviewer 2 Report
This manuscript describes the investigation of cytotoxicity, liver stage activity, gametocytocidal activity, beta-haematin inhibition activity under oxidizing and reducing conditions and in vivo activity in a P. berghei mouse model of an artemisinin-4-amino-7-chloroquinoline hybrid molecule previously synthesized and tested for in vitro activity against P. falciparum. Overall, the manuscript is well constructed and written and reports results that will be of interest to the malaria research community, especially given the multi-stage activity of this compound. Findings are well supported by the data. The only disappointment is the lack of activity of the hybrid in the mouse model, especially in view of the fact that the related drug combination shows good activity. It is a pity that the authors did not look at the metabolism of the hybrid in a microsomal model. It is quite possible that the ester is extensively metabolized in vivo to generate the 4-aminoquinoline 2 and artemisinin derivative 3. The former is almost 2 orders of magnitude less active than CQ, while the latter likely has the short half-life typical of artemisinins. This could account for the initial decrease in parasitemia, but lack of longer term efficacy. At least the authors should discuss this possibility.
There are also a number of small errors that need correction:
1. In the abstract, the Greek letter beta has been omitted from beta-hematin
2. In the caption to Figure 1 it is not correct to say that Fe(III)PPIX is dimerized to beta-hematin and then crystallized to hemozoin since the two are identical except that the latter is the natural, rather than synthetic product. The description in the Figure itself is correct.
3. In lines 188 and 192, the use of the word fundamental is not correct. This should rather be desirable or preferred.
4. In figure 4 the caption needs to explain clearly what the left and right axes refer to. This is explained in the text, but saying more in the caption will make it easier for the reader.
5. Line 214, the singular of species is also species, not specie. Specie means coins, especially gold coins.
6. All drugs such as chloroquine, artemisinin etc. should not be written with a capital letter.
Author Response

(The authors gave the same response as above.)

Reviewer 3 Report
Dear Editor,
Thank you! for inviting me to evaluate the article. Titled “Studies of potency and efficacy of an optimized dihydroarte- 2 misinin-quinoline hybrid against multiple stages of the Plas- 3 modium life cycle”. This paper was well written with all the details.
1) I suggest authors add figures in the Introduction part, helping the readers to understand the background of this work.
2) In figure 3, integrate the A and B were missing.
3) The letters in figure 4 were intensive, make more visible.
4) The references are not in the format, please make sure that all.
Author Response
First of all, we are thankful for the reviewer's work in revising our manuscript. We appreciate these comments. In this new manuscript version, the text was edited as pointed out above.